# Unveiling Shadows: Investigating women's experience of intimate partner violence in Ghana through the lens of the 2022 Demographic and Health Survey

Kwamena Sekyi Dickson[1], Castro Ayebeng[1], Joshua Okyere[1,2]*

1 Department of Population and Health, University of Cape Coast, Cape Coast, Ghana, 2 School of Nursing and Midwifery, College of Health Sciences, Kwame Nkrumah University of Science and Technology, Kumasi, Ghana

* joshuaokyere54@gmail.com

## Abstract

### Introduction

Intimate partner violence (IPV) is a significant public health issue, predominantly among women in sub-Saharan Africa, including Ghana. Existing evidence indicates high rates of IPV perpetration and its associated adverse health outcomes. Despite previous studies, reliance on old data underscores the need for current, nationally representative data to inform policy-making and interventions. Therefore, this study utilizes the 2022 Ghana Demographic and Health Survey (GDHS) to examine IPV experiences and associated factors, aiming to provide updated insights for effective IPV mitigation strategies in Ghana.

### Methods

This is a cross-sectional study of a weighted sample of 3,741 women between the ages of 15–49 years old from the 2022 GDHS. Binary logistic regression analysis was used in predicting the outcome.

### Results

The study found that 36.4 percent of women in Ghana had experienced some form of IPV, primarily emotional violence (31.5%), physical violence (17.3%), and sexual violence (7.6%). Women with higher levels of education had a significantly reduced risk of 60% of experiencing some form of IPV than those with no formal education. Partner domineering behaviours, such as women whose partners often get jealous for seeing them talk with other men [adjusted OR:1.76, 95%CI:1.25,2.48], accusing them of unfaithfulness [adjusted OR:2.59, 95%CI:1.03,2.46], not permitting them to meet female friends [adjusted OR:1.1.59, 95%CI:1.03,2.46], and limiting their contact with family [adjusted OR:5.75, 95%CI:2.27,13.42], were more likely to experience at least one form of IPV. Similarly, women who justified or endorsed wife beating had a higher likelihood [aOR = 1.57, 95%CI =

Third party data was obtained for this study from DHS Program. Data may be requested from DHS Program after creating an account and submitting a concept note. More access information can be found on the DHS Program website (https://dhsprogram.com/data/Access-Instructions.cfm). The authors confirm that interested researchers would be able to access these data in the same manner as the authors. The authors also confirm that they had no special access privileges that others would not have.

**Funding:** The author(s) received no specific funding for this work.

**Competing interests:** The authors have declared that no competing interests exist.

**Abbreviations:** AOR, Adjusted Odds Ratio; GDHS, Ghana Demographic and Health Survey; IPV, Intimate Partner Violence; Ref, Reference Category; SSA, Sub-Saharan Africa; WHO, World Health Organization.

1.22,2.02] of experiencing at least one form of IPV than those who did not endorse such behaviour.

## Conclusion

Identifying educational attainment, partner dominance, and violence endorsement as IPV predictors underscore targeted interventions. Promoting women's education bolsters empowerment and IPV prevention. Addressing dominance through education, counselling, and legal frameworks is crucial for fostering safer relationships and challenging violence normalization.

## Background

Intimate partner violence (IPV)–that is, any act perpetrated by an intimate partner to cause physical, emotional and sexual harm–is considered a serious public health concern and an infringement on human rights [1]. Globally, 1 in 3 women are victims of IPV [2]. In sub-Saharan Africa (SSA), approximately 33% of women experience IPV at some point in their lifetime [3]. These statistics highlight the urgency for all countries, particularly those located in SSA to work towards the elimination of IPV.

Evidence from Ghana [4] indicates that 28.2 percent of women justify their partners' perpetuation of IPV. This is worrying given that IPV is known to be associated with adverse physical, social and sexual health outcomes. For instance, women who experience sexual violence are 1.11 times more likely to engage in multiple high-risk fertility behaviours [5]. Furthermore, IPV perpetuation has been linked to higher risk of subsequent violence victimization, psychological distress, and unplanned pregnancies [6, 7]. One integrative review [8] has reported that women who experience IPV are more likely to experience cardiovascular symptoms.

Indeed, there is extant literature on IPV in Ghana. However, most of these studies were either based on the 2014 Ghana Demographic and Health Survey (GDHS) [4, 9, 10] or qualitative research approaches [11, 12]. However, given the significant time lapse between 2014 and 2022, relying solely on the findings of the 2014 GDHS may not accurately capture the current landscape of IPV experienced by women in Ghana. This underscores the critical need for more recent, nationally representative data to assess the prevalence of IPV, its various typologies, and the factors associated with it. Consequently, this study seeks to address this gap by utilizing the 2022 GDHS to examine women's experiences of IPV in Ghana and explore the factors contributing to it. By doing so, the study aims to provide up-to-date insights that can inform policy-making, intervention strategies, and support services aimed at addressing and mitigating IPV in Ghana.

## Methods

### Data source

**Data source and design.** The analysis utilized data from the 2022 Ghana Demographic and Health Survey (GDHS), specifically drawing from the individual recode file (IR). The GDHS is part of a larger initiative spanning 85 low-and-middle-income countries (LMICs) [13]. The primary objective of the 2022 GDHS was to furnish current estimations of fundamental demographic and health indicators. To achieve this goal, a meticulously designed two-staged sampling strategy was implemented, resulting in a stratified representative sample

comprising 18,450 households distributed across 618 clusters. This method ensured thorough representation at the national level, encompassing both urban and rural areas, and within each of Ghana's 16 regions [14].

During the initial stage, 618 target clusters were identified using a probability proportional to size method, considering urban and rural differentiations within each region [14]. Subsequently, an equal probability systematic random sampling approach was employed to choose the requisite number of clusters in both urban and rural settings. Moving to the second stage, subsequent to cluster selection, an exhaustive household listing and map updating process were conducted within all selected clusters, establishing a comprehensive roster of households for each cluster. From each cluster, 30 households were randomly chosen for interviews [14]. For further information on the design of the GDHS, refer to. https://www.dhsprogram.com/Methodology/Survey-Methodology.cfm.

Data collection was performed using structured household and women's questionnaires administered by trained enumerators. The women's questionnaire covered various aspects, including birth history, childhood mortality, fertility preferences, child health, maternal health, and domestic violence. The analysis consisted of a weighted sample of 3,741 women between the ages 15–49 years old who were either married or cohabiting and have information on intimate partner violence considered in this study. The use of the dataset did not require ethical clearance since it was obtained from a secondary data source. Nevertheless, permission to use the dataset was obtained from the Measure DHS.

## Study variables and measurements

**Outcome variable.** Intimate partner violence was derived from four outcome variables. These are experienced sexual violence ("yes" or "no."), emotional violence ("yes" or "no."), physical violence ("yes" or "no.") and experienced at least one form of violence ("yes" or "no.").

**Explanatory variables.** Multiple factors informed by theoretical and empirical literature relating to intimate partner violence [4, 11, 15, 16] were included in the analyses as the explanatory variables. These variables include age (15–19, 20–24, 25–29, 30–34, 35–39, 40–44, 45–49), region of residence (Western, Central, Greater Accra, Volta, Eastern, Ashanti, Western North, Ahafo, Bono, Bono East, Oti, Northern, Savannah, North East, Upper East, Upper West) residence (rural and urban), level of education (no education, basic, secondary and above), wealth index (poorest, poorer, middle, richer, richest), partner domineering variables include husband/partner are jealous if respondent talks with other men (never, often, sometimes, yes, but not in the last 12 months), husband/partner accuses respondent of unfaithfulness (never, often, sometimes, yes, but not in the last 12 months), husband/partner does not permit respondent to meet female friends (never, often, sometimes, yes, but not in the last 12 months), husband/partner tries to limit respondent's contact with family (never, often, sometimes, yes, but not in the last 12 months), husband/partner insists on knowing where respondent (never, often, sometimes, yes, but not in the last 12 months), and justification of wife beaten (no, yes).

## Data analysis

Statistical analysis was performed using Stata version 17. Both descriptive and inferential statistics were conducted. Further, univariate and multivariate analyses were conducted to explore the relationships between the explanatory variables and outcome variable. Logistic regression models were fitted using STATA regression commands to assess adjusted risk factors associated with the study outcomes, with Odds Ratios (OR) and 95% confidence intervals (CI) calculated. To account for any sampling bias from under or over-sampling of respondents

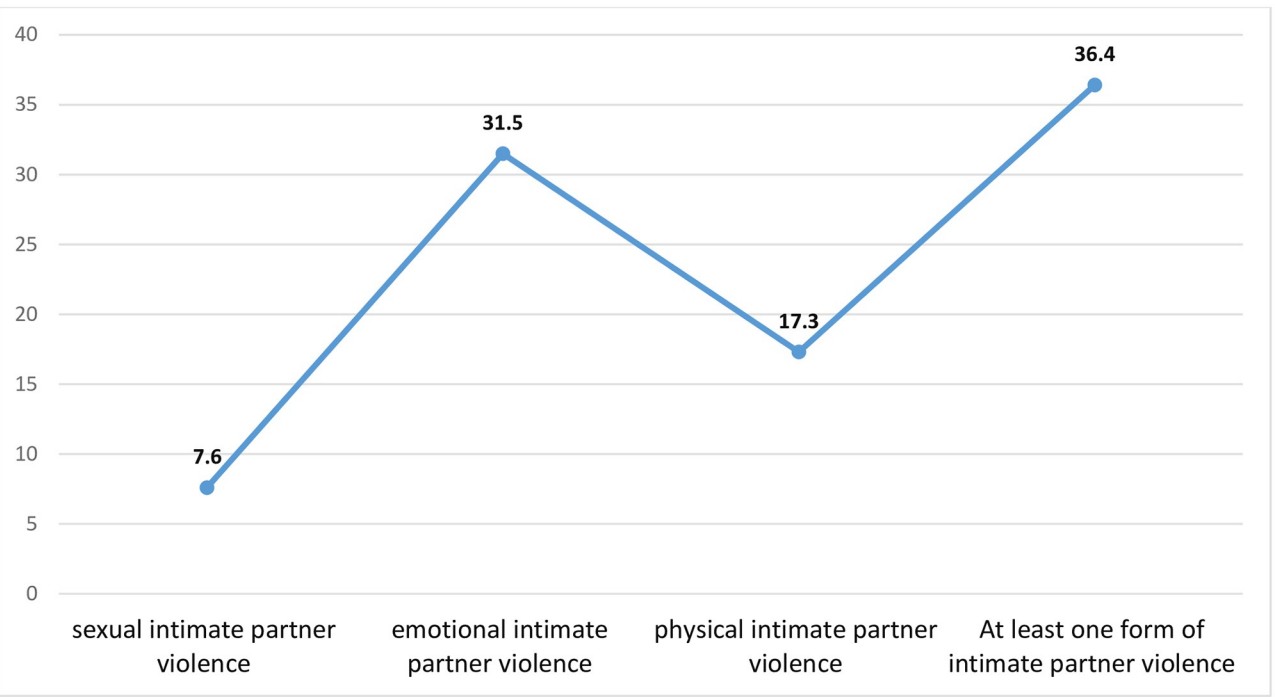

**Fig 1. Prevalence of intimate partner violence.**

in the total population, all descriptive estimates were weighted using the individual weight variable (d005) in the dataset. To consider the complex survey design of the DHS data, the "svyset" command in Stata was employed.

### Ethical approval and consent to participate

We did not need to seek ethical clearance because the DHS dataset we used is publicly available. We obtained the datasets from the DHS Program after completing the necessary registration and getting approval for their use. We followed all the ethical guidelines that pertain to using secondary datasets in research publications. You can find detailed information about how we used the DHS data and the ethical standards we followed at this link: http://goo.gl/ny8T6X.

## Results

### Prevalence of intimate partner violence

Fig 1 shows the prevalence of IPV among women in Ghana. It shows that 36.4 percent of women had experienced at least one form of IPV, with emotional IPV being the most prevalent (31.5%) followed by physical (17.3%) and sexual IPV (7.6%).

### Background characteristics and experienced intimate partner violence

Table 1 illustrates that IPV prevalence varies significantly across different demographic and behavioural characteristics. Regarding region of residence, women residing in the Savannah region show the highest IPV proportions of 15.3% sexual IPV, 46.1% emotional IPV, 27.0% physical IPV, and 56.7% experiencing at least one form of IPV. Conversely, the Bono region

**Table 1.** Background characteristics and intimate partner violence.

| Background characteristics | Frequency N = 3,741 | Proportion experienced sexual intimate partner violence | Proportion experienced emotional intimate partner violence | Proportion experienced physical intimate partner violence | Proportion experienced at least one form of intimate partner violence |
|---|---|---|---|---|---|
| **Demographic** | | | | | |
| *Age* | | $X^2 = 10.5$ (p = 0.106) | $X^2 = 10.7$ (p = 0.097) | $X^2 = 7.6$ (p = 0.269) | $X^2 = 8.7$ (p = 190) |
| 15–19 | 85 | 3.7 | 23.8 | 23.5 | 34.1 |
| 20–24 | 499 | 8.2 | 32.9 | 20.3 | 36.7 |
| 25–29 | 674 | 7.4 | 29.2 | 13.6 | 34.5 |
| 30–34 | 792 | 6.7 | 30.6 | 15.7 | 35.0 |
| 35–39 | 743 | 9.4 | 26.8 | 15.1 | 33.5 |
| 40–44 | 587 | 8.3 | 37.8 | 21.1 | 41.4 |
| 45–49 | 411 | 5.6 | 36.4 | 20.4 | 40.2 |
| **Region of residence** | | $X^2 = 48.7$ (p<0.001) | $X^2 = 94.2$ (p<0.001) | $X^2 = 58.1$ (p<0.001) | $X^2 = 113.7$ (p<0.001) |
| Western | 205 | 7.6 | 38.0 | 18.8 | 42.4 |
| Central | 376 | 10.1 | 40.0 | 23.0 | 47.0 |
| Greater Accra | 536 | 7.1 | 29.6 | 10.6 | 30.4 |
| Volta | 164 | 9.7 | 35.0 | 14.6 | 37.6 |
| Eastern | 285 | 6.5 | 31.2 | 19.0 | 34.9 |
| Ashanti | 722 | 9.6 | 28.0 | 15.6 | 34.6 |
| Western north | 100 | 5.7 | 22.4 | 12.6 | 25.9 |
| Ahafo | 80 | 8.5 | 38.4 | 21.9 | 48.1 |
| Bono | 127 | 2.0 | 21.7 | 13.4 | 25.3 |
| Bono east | 163 | 5.7 | 26.4 | 15.8 | 30.6 |
| Oti | 105 | 11.2 | 36.2 | 21.4 | 42.4 |
| Northern | 384 | 5.7 | 34.3 | 23.2 | 40.7 |
| Savannah | 95 | 15.3 | 46.1 | 27.0 | 56.7 |
| North east | 93 | 3.5 | 25.1 | 16.0 | 28.4 |
| Upper east | 193 | 3.3 | 29.1 | 17.3 | 32.8 |
| Upper west | 112 | 6.9 | 22.8 | 13.4 | 28.6 |
| *Place of residence* | | $X^2 = 1.2$ (p = 0.278) | $X^2 = 2.3$ (p = 0.129) | $X^2 = 4.5$ (p<0.05) | $X^2 = 3.6$ (p = 0.058) |
| Urban | 1,962 | 8.2 | 30.2 | 15.7 | 34.7 |
| Rural | 1,779 | 7.0 | 32.9 | 19.0 | 38.3 |
| *Education* | | $X^2 = 5.0$ (p = 0.169) | $X^2 = 40.4$ (p<0.001) | $X^2 = 46.8$ (p<0.001) | $X^2 = 38.7$ (p<0.001) |
| No education | 927 | 8.4 | 35.1 | 23.5 | 41.7 |
| Primary | 565 | 8.8 | 40.7 | 20.8 | 44.4 |
| Secondary | 1,900 | 7.5 | 29.9 | 15.5 | 34.6 |
| Higher | 349 | 4.1 | 15.5 | 4.7 | 19.2 |
| *Wealth status* | | $X^2 = 8.3$ (p = 0.080) | $X^2 = 14.5$ (p<0.01) | $X^2 = 40.7$ (p<0.001) | $X^2 = 19.5$ (p<0.01) |
| Poorest | 739 | 6.2 | 33.7 | 21.5 | 40.0 |
| Poorer | 685 | 7.2 | 35.0 | 17.5 | 39.3 |
| Middle | 717 | 9.0 | 35.7 | 20.6 | 41.3 |
| Richer | 816 | 9.9 | 30.9 | 20.0 | 36.1 |
| Richest | 784 | 5.6 | 23.2 | 7.3 | 26.1 |
| **Marita status** | | $X^2 = 29.4$ (p<0.001) | $X^2 = 15.8$ (p<0.001) | $X^2 = 6.0$ (p<0.05) | $X^2 = 11.6$ (p<0.01) |
| Married | 2,700 | 6.4 | 29.2 | 15.9 | 34.2 |
| Cohabitation | 1,041 | 10.9 | 37.6 | 20.9 | 42.1 |
| **Occupation** | | $X^2 = 14.8$ (p = 0.094) | $X^2 = 30.1$ (p<0.001) | $X^2 = 26.9$ (p<0.001) | $X^2 = 32.9$ (p<0.001) |
| Not working | 394 | 9.9 | 27.9 | 17.1 | 34.2 |
| Professional/technical | 248 | 3.5 | 20.3 | 4.5 | 24.6 |

*(Continued)*

**Table 1.** (Continued)

| Background characteristics | Frequency N = 3,741 | Proportion experienced sexual intimate partner violence | Proportion experienced emotional intimate partner violence | Proportion experienced physical intimate partner violence | Proportion experienced at least one form of intimate partner violence |
|---|---|---|---|---|---|
| Clerical | 49 | 4.0 | 14.5 | 1.3 | 15.1 |
| Sales | 360 | 8.7 | 31.4 | 16.6 | 34.9 |
| Agriculture–self employed | 5 | 0.0 | 23.5 | 7.8 | 27.5 |
| Agriculture—employee | 148 | 4.8 | 28.6 | 16.5 | 32.2 |
| Services | 2,110 | 7.9 | 34.9 | 19.7 | 40.2 |
| Skilled manual | 392 | 6.5 | 27.5 | 16.0 | 31.6 |
| Unskilled manual | 17 | 8.6 | 22.9 | 13.5 | 24.6 |
| Other | 18 | 20.5 | 38.9 | 8.9 | 44.1 |
| **Partner Domineering** | | | | | |
| *Husband/partner jealous if respondent talks with other men* | | X² = 156.0 (p<0.001) | X² = 325.3 (p<0.001) | X² = 262.8 (p<0.001) | X² = 326.0 (p<0.001) |
| Never | 2,340 | 3.9 | 21.4 | 9.8 | 25.8 |
| Often | 514 | 20.3 | 57.1 | 37.6 | 62.5 |
| Sometimes | 771 | 9.7 | 42.5 | 23.6 | 47.7 |
| Yes, but not in the last 12 months | 116 | 12.2 | 48.7 | 35.9 | 58.8 |
| *Husband/partner accuses respondent of unfaithfulness* | | X² = 191.2 (p<0.001) | X² = 370.0 (p<0.001) | X² = 250.9 (p<0.001) | X² = 338 (p<0.001) |
| Never | 3,079 | 5.1 | 24.2 | 12.6 | 29.4 |
| Often | 247 | 27.0 | 70.7 | 45.9 | 73.2 |
| Sometimes | 350 | 14.8 | 60.3 | 32.4 | 64.8 |
| Yes, but not in the last 12 months | 65 | 16.8 | 70.8 | 47.0 | 75.0 |
| *Husband/partner does not permit respondent to meet female friends* | | X² = 127.7 (p<0.001) | X² = 208.7 (p<0.001) | X² = 141.5 (p<0.001) | X² = 207.0 (p<0.001) |
| Never | 3,143 | 5.6 | 26.9 | 14.1 | 31.6 |
| Often | 269 | 21.7 | 59.0 | 36.3 | 62.8 |
| Sometimes | 285 | 15.2 | 55.9 | 32.9 | 64.4 |
| Yes, but not in the last 12 months | 44 | 18.9 | 33.3 | 24.1 | 36.7 |
| *Husband/partner tries to limit respondent's contact with family* | | X² = 156.2 (p<0.001) | X² = 154.1 (p<0.001) | X² = 171.3 (p<0.001) | X² = 154.4 (p<0.001) |
| Never | 3,499 | 6.3 | 28.8 | 15.2 | 33.6 |
| Often | 84 | 34.4 | 80.9 | 57.8 | 88.4 |
| Sometimes | 147 | 22.9 | 63.5 | 39.9 | 69.9 |
| Yes, but not in the last 12 months | 11 | 14.2 | 69.0 | 53.9 | 76.3 |
| *Husband/partner insists on knowing where respondent is* | | X² = 115.1 (p<0.001) | X² = 250.0 (p<0.001) | X² = 168.7 (p<0.001) | X² = 282.9 (p<0.001) |
| Never | 2,189 | 3.9 | 22.3 | 10.5 | 26.2 |
| Often | 880 | 15.9 | 48.3 | 29.9 | 54.8 |
| Sometimes | 624 | 8.7 | 39.5 | 22.8 | 45.7 |
| Yes, but not in the last 12 months | 48 | 9.7 | 37.4 | 25.0 | 42.1 |
| **Justification of wife beaten** | | X² = 7.1 (p<0.01) | X² = 41.7 (p<0.001) | X² = 113.4 (p<0.001) | X² = 74.5 (p<0.001) |
| No | 2,958 | 7.2 | 29.1 | 13.8 | 33.0 |

*(Continued)*

**Table 1.** (Continued)

| Background characteristics | Frequency N = 3,741 | Proportion experienced sexual intimate partner violence | Proportion experienced emotional intimate partner violence | Proportion experienced physical intimate partner violence | Proportion experienced at least one form of intimate partner violence |
|---|---|---|---|---|---|
| Yes | 783 | 9.2 | 40.4 | 30.2 | 49.1 |
| Total | 3,741 | 7.6 | 31.5 | 17.3 | 36.4 |

has the lowest proportions, with 2.0% sexual IPV, 21.7% emotional IPV, 13.4% physical IPV, and 25.3% facing any form of IPV.

Education level also plays a role, with women lacking formal education exhibiting the highest proportions of 8.4% sexual IPV, 35.1% emotional IPV, 23.5% physical IPV, and 41.7% experiencing at least one of the forms of IPV. However, those with higher education levels report the lowest proportions of 4.1% sexual IPV, 15.5% emotional IPV, 4.7% physical IPV, and 19.2% facing any form of IPV.

Marital status is another significant factor, with women in cohabiting relationships showing the highest proportions, with 10.9% reporting sexual IPV, 37.6% emotional IPV, 20.9% physical IPV, and 42.1% experiencing at least one form of IPV. Contrariwise, married individuals report the lowest proportions of 6.4% sexual IPV, 29.2% emotional IPV, 15.9% physical IPV, and 34.2% facing at least one form of IPV.

Furthermore, women whose partners exhibit domineering behaviour significantly show a higher prevalence of IPV. For instance, women who were often accused of unfaithfulness exhibit the highest proportions of IPV, with 27.0% sexual IPV, 70.7% emotional IPV, 45.9% physical IPV, and 73.2% facing any form of IPV compared to their counterparts who have never experienced these domineering attitudes from their partners.

## Binary logistic regression analysis of the experience of intimate partner violence

Table 2 shows the binary logistic regression results of the factors associated with different forms of IPV. The results show that region of residence, level of education, partner domineering behaviour, and justification of violence were significantly associated with the experience of IPV. Women who were residing in the Northeast [aOR = 0.35, 95%CI = 0.17,0.71], Bono [aOR = 0.36, 95%CI = 0.19,0.65], Upper West [aOR = 0.33, 95%CI = 0.19,0.56], and Western North [aOR = 0.35, 95%CI = 0.17,0.71] regions had lower odds experiencing any form of IPV compared to their counterparts in the Western region.

The results demonstrate that increasing levels of women's education have a protective effect on their likelihood of experiencing the three forms of IPV considered in this study. For instance, women with higher levels of education had a reduced risk of 60% of experiencing at least one form of IPV than those with no formal education.

However, women whose partners exhibit domineering behaviours were at a higher risk of experiencing IPV. Thus, women whose partners often get jealous for seeing them talk with other men, accusing them of unfaithfulness, not permitting them to meet female friends, and limiting their contact with family were 1.76, 2.59, 1.59, and 5.75 times, respectively, more likely to experience at least one form of IPV. Similarly, women who justified or endorsed wife beating were more had a higher likelihood [aOR = 1.57, 95%CI = 1.22,2.02] of experiencing at least one form of IPV than those who did not endorse such behaviour.

**Table 2. Binary logistic regression and experienced of intimate partner violence.**

| Background characteristics | Sexual intimate partner violence | Emotional intimate partner violence | Physical intimate partner violence | At least one form of intimate partner violence |
|---|---|---|---|---|
| | A0R (CI) | A0R (CI) | A0R (CI) | A0R (CI) |
| **Demographic** | | | | |
| *Age* | | | | |
| 15–19 | Ref | Ref | Ref | Ref |
| 20–24 | 2.59(0.90, 7.43) | 1.37(0.63, 2.99) | 0.70(0.22, 2.23) | 0.90 (0.38, 2.17) |
| 25–29 | 2.69(0.93, 7.75) | 1.34(0.62, 2.94) | 0.45(0.15, 1.35) | 0.98(0.41, 2.33) |
| 30–34 | 2.66(0.90, 7.93) | 1.75(0.81, 3.79) | 0.65(0.23, 1.84) | 1.19(0.51, 2.76) |
| 35–39 | 3.95**(1.43, 10.88) | 1.45(0.66, 3.19) | 0.64(0.21, 1.95) | 1.16(0.48, 2.81) |
| 40–44 | 3.43*(1.16, 10.18) | 2.41*(1.06, 5.48) | 0.92(0.31, 2.70) | 1.53(0.64, 3.70) |
| 45–49 | 2.23(0.68, 7.30) | 2.43*(1.01, 5.87) | 0.93(0.31, 2.83) | 1.54(0.59, 4.01) |
| **Region of residence** | | | | |
| Western | Ref | Ref | Ref | Ref |
| Central | 0.80(0.35, 1.81) | 0.65(0.37, 1.13) | 0.68(0.34, 1.33) | 0.75(0.44, 1.28) |
| Greater Accra | 0.94(0.45, 1.96) | 0.86(0.51, 1.43) | 0.61(0.31, 1.18) | 0.71(0.44, 1.16) |
| Volta | 1.01(0.47, 2.18) | 0.61(0.36, 1.04) | 0.52*(0.28, 0.98) | 0.57**(0.35, 0.94) |
| Eastern | 0.74(0.33, 1.67) | 0.57*(0.33, 0.99) | 0.87(0.49, 1.56) | 0.60*(0.36, 0.98) |
| Ashanti | 1.11(0.49, 2.49) | 0.45**(0.36, 0.77) | 0.54*(0.29, 1.00) | 0.54**(0.33, 0.90) |
| Western north | 0.85(0.32, 2.21) | 0.39**(0.22, 0.69) | 0.48*(0.24, 0.97) | 0.39**(0.22, 0.69) |
| Ahafo | 1.07(0.48, 2.38) | 0.83(0.44, 1.54) | 0.91(0.51, 1.63) | 1.00(0.58, 1.72) |
| Bono | 0.21*(0.05, 0.78) | 0.35**(0.18, 0.68) | 0.61(0.28, 1.32) | 0.36**(0.19, 0.65) |
| Bono east | 0.91(0.35, 2.37) | 0.59(0.34, 1.03) | 0.80(0.41, 1.56) | 0.59(0.34, 1.02) |
| Oti | 1.47(0.68, 3.16) | 0.62(0.34, 1.07) | 0.74(0.39, 1.38) | 0.64(0.39, 1.07) |
| Northern | 0.57(0.23, 1.43) | 0.61(0.33, 1.12) | 0.65(0.36, 1.19) | 0.60(0.50, 1.77) |
| Savannah | 1.44(0.50, 4.17) | 0.78(0.41, 1.49) | 0.57(0.31, 1.05) | 0.95(0.50, 1.77) |
| North east | 0.41(0.15, 1.09) | 0.40*(0.19, 0.84) | 0.44(0.19, 1.03) | 0.35**(0.17, 0.71) |
| Upper east | 0.50(0.16, 1.55) | 0.63(0.31, 1.27) | 0.84(0.42, 1.64) | 0.59(0.29, 1.19) |
| Upper west | 0.78(0.34, 1.79) | 0.32***(0.18, 0.57) | 0.33**(0.16, 0.68) | 0.33***(0.19, 0.56) |
| *Place of residence* | | | | |
| Urban | Ref | Ref | Ref | Ref |
| Rural | 0.91(0.59, 1.41) | 1.00(0.74, 1.35) | 1.01(0.71, 1.46) | 0.99(0.75, 1.31) |
| *Education* | | | | |
| No education | Ref | Ref | Ref | Ref |
| Primary | 0.70(0.38, 1.28) | 1.11(0.69, 1.45) | 0.75(0.50, 1.12) | 1.01(0.71, 1.44) |
| Secondary | 0.71(0.39, 1.31) | 0.85(0.60, 1.20) | 0.72(0.50, 1.03) | 0.85(0.57, 1.20) |
| Higher | 0.72(0.28, 1.81) | 0.39**(0.19, 0.78) | 0.51(0.19, 1.35) | 0.40**(0.21, 0.77) |
| *Wealth status* | | | | |
| Poorest | Ref | Ref | Ref | Ref |
| Poorer | 0.86(0.49, 1.50) | 1.00(0.69, 1.45) | 0.71(0.49, 1.03) | 1.13(0.86, 1.48) |
| Middle | 1.18(0.65, 2.15) | 0.99(0.65, 1.50) | 0.93(0.56, 1.53) | 0.97(0.64, 1.45) |
| Richer | 1.51(0.83, 2.76) | 0.91(0.56, 1.47) | 1.29(0.76, 2.19) | 0.91(0.58, 1.44) |
| Richest | 1.09(0.49, 2.45) | 0.75(0.41, 1.34) | 0.54(0.27, 1.11) | 0.69(0.39, 1.21) |
| **Marita status** | | | | |
| Married | Ref | Ref | Ref | Ref |
| Cohabitation | 1.27(0.83, 1.97) | 1.22(0.91, 1.63) | 1.13(0.79, 1.60) | 1.13(0.86, 1.48) |
| **Occupation** | | | | |
| Not working | Ref | Ref | Ref | Ref |
| Professional/technical | 0.35*(0.13, 0.98) | 1.70(0.84, 3.44) | 0.54(0.21, 1.39) | 1.52(0.80, 2.93) |
| Clerical | 0.45(0.14, 1.28) | 0.86(0.30, 2.41) | 0.12*(0.01, 0.98) | 0.61(0.23, 1.66) |

*(Continued)*

**Table 2.** (Continued)

| Background characteristics | Sexual intimate partner violence | Emotional intimate partner violence | Physical intimate partner violence | At least one form of intimate partner violence |
|---|---|---|---|---|
| | A0R (CI) | A0R (CI) | A0R (CI) | A0R (CI) |
| Sales | 0.75(0.35, 1.59) | 1.01(0.62, 1.64) | 0.99(0.56, 1.74) | 0.94(0.59, 1.50) |
| Agriculture–self employed | 1 | 0.44(0.08, 2.47) | 0.32(0.05, 1.92) | 0.37(0.19, 1.61) |
| Agriculture—employee | 0.48(0.18, 1.32) | 0.83(0.50, 1.42) | 0.69(0.37, 1.27) | 0.72(0.43, 1.18) |
| Services | 0.66(0.36, 1.19) | 1.27(0.88, 1.84) | 1.21(0.78, 1.89) | 1.21(0.85, 1.71) |
| Skilled manual | 0.59(0.29, 1.24) | 1.08(0.67, 1.76) | 1.14(0.62, 2.09) | 0.95(0.59, 1.54) |
| Unskilled manual | 1.04(0.18, 6.10) | 0.82(0.33, 2.04) | 0.89(0.15, 5.29) | 0.64(0.28, 1.50) |
| Other | 3.56(0.68, 18.48) | 1.67(0.40, 7.04) | 0.54(0.10, 2.73) | 1.69(0.42, 6.84) |
| **Partner Domineering** | | | | |
| *Husband/partner jealous if respondent talks with other men* | | | | |
| Never | Ref | Ref | Ref | Ref |
| Often | 1.94(1.00, 3.76) | 1.80**(1.28, 2.54) | 2.46***(1.62, 3.73) | 1.76**(1.25, 2.48) |
| Sometimes | 1.69(0.96, 3.00) | 1.72***(1.31, 2.26) | 1.94***(1.37, 2.75) | 1.66***(1.29, 2.13) |
| Yes, but not in the last 12 months | 2.37(0.81, 7.00) | 2.47**(1.27, 4.80) | 4.37**(1.89, 10.12) | 3.20***(1.71, 5.98) |
| *Husband/partner accuses respondent of unfaithfulness* | | | | |
| Never | Ref | Ref | Ref | Ref |
| Often | 2.21*(1.04, 4.72) | 3.23***(1.91, 5.45) | 2.11**(1.19, 3.74) | 2.59***(1.03, 2.46) |
| Sometimes | 1.60(0.86, 2.97) | 2.70***(1.96, 3.73) | 1.62*(1.10, 2.38) | 2.42***(1.22, 2.97) |
| Yes, but not in the last 12 months | 2.29(0.82, 6.38) | 5.53***(2.45, 12.52) | 3.40(0.99, 11.66) | 4.93***(2.18, 11.42) |
| *Husband/partner does not permit respondent to meet female friends* | | | | |
| Never | Ref | Ref | Ref | Ref |
| Often | 1.60(0.85, 3.03) | 1.75*(1.13, 2.72) | 1.41(0.88, 2.26) | 1.59*(1.03, 2.46) |
| Sometimes | 1.36(0.73, 2.52) | 1.62*(1.01, 2.61) | 1.24(0.79, 1.95) | 1.90**(1.22, 2.97) |
| Yes, but not in the last 12 months | 1.97(0.78, 4.97) | 0.55(0.21, 1.41) | 0.73(0.28, 1.88) | 0.45(0.18, 1.12) |
| *Husband/partner tries to limit respondent's contact with family* | | | | |
| Never | Ref | Ref | Ref | Ref |
| Often | 2.73*(1.24, 6.02) | 3.56**(1.60, 7.92) | 2.67*(1.14, 6.27) | 5.75***(2.27, 13.42) |
| Sometimes | 2.55*(1.22, 5.34) | 2.13**(1.27, 3.58) | 1.97*(1.10, 3.53) | 2.32**(1.35, 3.99) |
| Yes, but not in the last 12 months | 1.46(0.34, 6.37) | 2.52(0.69, 9.12) | 2.91(0.69, 12.28) | 3.01(0.90, 10.07) |
| *Husband/partner insists on knowing where respondent is* | | | | |
| Never | Ref | Ref | Ref | Ref |
| Often | 1.98**(1.20, 3.27) | 1.55**(1.60, 7.92) | 1.78**(1.22, 2.58) | 1.70***(1.27, 2.29) |
| Sometimes | 1.37(0.83, 2.27) | 1.34*(1.02, 1.76) | 1.40(0.96, 2.04) | 1.41**(1.07, 1.86) |
| Yes, but not in the last 12 months | 0.77(0.25, 2.34) | 1.25(0.54, 2.87) | 1.18(0.52, 2.65) | 1.19(0.53, 2.70) |
| **Justification of wife beaten** | | | | |
| No | Ref | Ref | Ref | Ref |
| Yes | 1.12(0.68, 1.84) | 1.33*(1.00, 1.77) | 2.31***(1.66, 3.20) | 1.57***(1.22, 2.02) |

***p<0.001,

**p<0.01,

*p<0.05

Ref = Reference category

A0R = Adjusted Odds Ratio, CI = Confidence interval

## Discussion

This study aims to provide current insight into women's experience of IPV in Ghana. Consistent with the global [1] and regional [2] estimates, we found that 36.4 percent of women had ever experienced one form of IPV. Additionally, our result aligns with previous studies from Pakistan [17] and Kenya [18] that have found emotional violence to be the most common IPV type experienced. The observed pattern of the IPV typologies is expected. Sexual violence often comes with heightened fear and shame which tends to discourage victims from reporting such acts. This assertion is corroborated by Orchowski et al. [19] who argue that victims of sexual violence do not report due to factors such as *"labeling of the experience, age, fear, privacy concerns, self-blame, betrayal/shock, the relation/power of the perpetrator, negative reactions to disclosure, and the belief—or personal experience—that reporting would not result in justice"*. As such, the low prevalence of sexual violence reported in this study may not be the true reflection of the situation but rather a reflection of the inability of women to open up about their experiences.

Regarding the associated factors, our study revealed that educational attainment, partner domineering behaviour and justification of violence were the only factors that significantly predicted the women's likelihood of experiencing at least one form of IPV. Specifically, women with higher educational attainment had a significantly lower likelihood of being victims of IPV. The result aligns with Iqbal and Fatmi [17] who found low educational attainment to be a high risk factor for IPV victimization. Our finding is corroborated by Weitzman [20] study which revealed that increasing women's schooling significantly reduces their risk of becoming victims of emotional, physical and sexual IPV. This association is best explained from the social causation perspective which postulates that an individual's socioeconomic status is the cause of their weakened mental health [21]. In the context of IPV, it implies that women with higher levels of education may have better access to resources, social support networks, and coping strategies, which in turn reduce their vulnerability to IPV. Furthermore, education is often linked to higher social status and exposure to progressive attitudes toward gender roles and interpersonal relationships. Women with higher education levels may be less likely to adhere to traditional gender norms that justify or tolerate violence against women. They may have more egalitarian relationships characterized by mutual respect and non-violent conflict resolution strategies.

Our study shows that women whose partners exhibited domineering attitudes (i.e., getting jealous, not permitting women to meet female friends, partner limiting woman's contact with her family, etc.) were more likely to experience at least one form of IPV. This association resonates with Capaldi et al. [22] who mapped literature and found possessiveness and jealousy on the part of male partners exacerbated the risk of IPV among women. According to Aizpurua et al. [23], partner domineering or controlling behaviours tend to generate fear in women and adversely impact their self-esteem. This diminished self-esteem may impede women from recognizing and addressing subtle forms of violence, which can escalate into more severe instances of IPV. Therefore, the association observed in our study underscores the importance of addressing and intervening in domineering behaviours within intimate relationships to mitigate the risk of IPV and safeguard the well-being of women.

We found that women who justified or endorsed wife beating were at a 57 percent higher risk of becoming victims of at least one form of IPV. Our result is in contrast to a Nepali study [24] that found no significant association between acceptance of wife beating and life-time or past year physical IPV experience among women. A possible explanation for this association could be that women who endorse such attitudes may be more likely to accept and internalize

abusive behaviours within their relationships, creating a conducive environment for IPV to occur. There were also significant differences in the risk of IPV by region; thus, underscoring a need for more region-specific tailored interventions to address IPV among women.

## Strengths and limitations

We relied on the 2022 GDHS which provides a closer current picture of IPV among women of reproductive age in Ghana. This ensures that our findings reflect the present status quo. Also, the GDHS data collection follows systematic methodologies and a sampling process which ensures that the findings are representative at the district and national levels. Nonetheless, the estimated prevalence of IPV may be underestimated given that the data is self-reported. Also, we are unable to establish a causal association between educational attainment, partner domineering behaviour, and the endorsement of violence as significant predictors of IPV since the GDHS is based on a cross-sectional design.

## Conclusion

In conclusion, the study's identification of educational attainment, partner domineering behaviour, and the endorsement of violence as significant predictors of IPV highlights areas for targeted intervention. Policies aimed at promoting education, particularly for women, not only contribute to their economic empowerment but also serve as a protective factor against IPV. Furthermore, addressing partner domineering behaviour through educational campaigns, counselling services, and legal frameworks can help empower women and create safer relationship dynamics. Efforts to challenge cultural attitudes that normalize or justify violence, such as the endorsement of wife beating, are essential for shifting societal norms and fostering environments where IPV is not tolerated.

## Acknowledgments

We acknowledge the Measure DHS for granting us free access to the dataset used in this study.

## Author Contributions

**Conceptualization:** Kwamena Sekyi Dickson, Castro Ayebeng, Joshua Okyere.

**Data curation:** Kwamena Sekyi Dickson.

**Formal analysis:** Kwamena Sekyi Dickson.

**Methodology:** Kwamena Sekyi Dickson, Castro Ayebeng, Joshua Okyere.

**Software:** Kwamena Sekyi Dickson.

**Supervision:** Castro Ayebeng, Joshua Okyere.

**Validation:** Castro Ayebeng, Joshua Okyere.

**Writing – original draft:** Kwamena Sekyi Dickson, Castro Ayebeng, Joshua Okyere.

**Writing – review & editing:** Kwamena Sekyi Dickson, Castro Ayebeng, Joshua Okyere.

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
