## [Decision Letter · Decision Letter 0]

3 Jun 2024

PONE-D-24-14998Unveiling Shadows: Investigating women’s experience of intimate partner violence in Ghana through the lens of the 2022 Demographic and Health SurveyPLOS ONE

Dear Dr. Okyere,

Thank you for submitting your manuscript to PLOS ONE. After careful consideration, we feel that it has merit but does not fully meet PLOS ONE’s publication criteria as it currently stands. Therefore, we invite you to submit a revised version of the manuscript that addresses the points raised during the review process.

We look forward to receiving your revised manuscript.

Kind regards,

Tijani Idris Ahmad Oseni, FMCFM

Academic Editor

PLOS ONE

Journal Requirements:

Additional Editor Comments:

Kindly address the issues raised by the reviewers and resubmit the manuscript for further evaluation.

Reviewers' comments:

Reviewer's Responses to Questions

**Comments to the Author**

1. Is the manuscript technically sound, and do the data support the conclusions?

Reviewer #1: No

Reviewer #2: Yes

2. Has the statistical analysis been performed appropriately and rigorously? 

Reviewer #1: No

Reviewer #2: Yes

3. Have the authors made all data underlying the findings in their manuscript fully available?

Reviewer #1: Yes

Reviewer #2: Yes

4. Is the manuscript presented in an intelligible fashion and written in standard English?

Reviewer #1: Yes

Reviewer #2: Yes

5. Review Comments to the Author

Reviewer #1: The study title was highly suggestive of a different form of study. The study design could not be applied to meet the study aims and answer the question of the research. The methods of the study were not described. The conclusions are not reflective of the study.

Reviewer #2: The research is about a significant public health issue and has identified predictors of IPV which could be addressed by the policy makers, therefore, recommended for publication.

The data used, analysis, result presentations, discussion and the conclusion are aligned. However, the author should clarify and look into the following observations:

1. Table 1, under the "demographic" (proportion of at least one form of IPV), the p value of 190 seems to be invalid.

2. Table 1, under "age group", the summation is 3,791 which is greater than "N" (3,741)

3. Table 1, under "region of residence" the summation is 3,740 which is lesser than "N" (3,741)

4. Table 1, the entire last row (Total) is unnecessary and seems to be confusing.

5. Table 2, the confidence interval (CI) should be presented as an "interval" (lower limit-upper limit), NOT with a "comma"

6. Table 2, it will be good if the p values are also included in the table, to show the EXACT values especially the significant ones. There may be need for more explanation than asterisks at the footnote.

7. The manuscript has been written in standard English, with very few typographical errors;

a) Under "binary logistic regression analysis", Paragraph 3, line 6, ".......beating WERE MORE had a higher likelihood..."

b) Under Table 2, "MARITA Status"

6. PLOS authors have the option to publish the peer review history of their article (what does this mean?). If published, this will include your full peer review and any attached files.

Reviewer #1: No

Reviewer #2: **Yes: **Dr Tawakalit Olubukola SALAM

---

## [Author Response · Author response to Decision Letter 0]

10 Jul 2024

RE PONE-D-24-14998

Reviewer #1: The study title was highly suggestive of a different form of study. The study design could not be applied to meet the study aims and answer the question of the research. The methods of the study were not described. The conclusions are not reflective of the study.

Response: Thank you for the comment and the time spent to critically review our manuscript. We have now changed the title to reflect the objectives and design of the study. The methods section has been properly described. The conclusion section has been revised to reflect the findings: “In conclusion, this study underscores the multifaceted nature of IPV in Ghana and the importance of addressing both individual and societal factors in efforts to reduce its prevalence. Educational empowerment, challenging harmful societal norms, and addressing controlling behaviors in intimate relationships are key strategies that must be prioritized. Efforts to challenge cultural attitudes that normalize or justify violence, such as the endorsement of wife beating, are essential for shifting societal norms and fostering environments where IPV is not tolerated.”

Reviewer #2: The research is about a significant public health issue and has identified predictors of IPV which could be addressed by the policy makers, therefore, recommended for publication.

Response: Thank you. We appreciate the time spent in reviewing our paper. 

The data used, analysis, result presentations, discussion and the conclusion are aligned. However, the author should clarify and look into the following observations:

1. Table 1, under the "demographic" (proportion of at least one form of IPV), the p value of 190 seems to be invalid.

Response: Thank you for drawing our attention. It is ‘0.190’. This correction has been made

2. Table 1, under "age group", the summation is 3,791 which is greater than "N" (3,741)

Response: This has been corrected

3. Table 1, under "region of residence" the summation is 3,740 which is lesser than "N" (3,741)

Response: This has been corrected

4. Table 1, the entire last row (Total) is unnecessary and seems to be confusing.

Response: We agree with you as that result had already been presented in Figure 1. Consequently, we have removed the last row (Total). 

5. Table 2, the confidence interval (CI) should be presented as an "interval" (lower limit-upper limit), NOT with a "comma"

Response: we have replaced all the "comma" with an "interval"

6. Table 2, it will be good if the p values are also included in the table, to show the EXACT values especially the significant ones. There may be need for more explanation than asterisks at the footnote.

Response: The EXACT values has been added

7. The manuscript has been written in standard English, with very few typographical errors;

a) Under "binary logistic regression analysis", Paragraph 3, line 6, ".......beating WERE MORE had a higher likelihood..."

Response: Thank you. We have revised this: “…beating had a higher likelihood”

b) Under Table 2, "MARITA Status"

Response: We have changed this to marital status.

---

## [Decision Letter · Decision Letter 1]

1 Aug 2024

Women’s experiences of intimate partner violence in Ghana: Update from the 2022 Demographic and Health Survey

PONE-D-24-14998R1

Dear Dr. Okyere,

We’re pleased to inform you that your manuscript has been judged scientifically suitable for publication and will be formally accepted for publication once it meets all outstanding technical requirements.

Kind regards,

Tijani Idris Ahmad Oseni, FMCFM

Academic Editor

PLOS ONE

Additional Editor Comments (optional):

Reviewers' comments:

Reviewer's Responses to Questions

**Comments to the Author**

1. If the authors have adequately addressed your comments raised in a previous round of review and you feel that this manuscript is now acceptable for publication, you may indicate that here to bypass the “Comments to the Author” section, enter your conflict of interest statement in the “Confidential to Editor” section, and submit your "Accept" recommendation.

Reviewer #1: All comments have been addressed

Reviewer #2: All comments have been addressed

2. Is the manuscript technically sound, and do the data support the conclusions?

Reviewer #1: Yes

Reviewer #2: Yes

3. Has the statistical analysis been performed appropriately and rigorously? 

Reviewer #1: Yes

Reviewer #2: Yes

4. Have the authors made all data underlying the findings in their manuscript fully available?

Reviewer #1: Yes

Reviewer #2: Yes

5. Is the manuscript presented in an intelligible fashion and written in standard English?

Reviewer #1: Yes

Reviewer #2: Yes

6. Review Comments to the Author

Reviewer #1: Authors have reviewed the article and corrections are acceptable for publication as it now is. The title is highly reflective of the content and the methods addresses the research question.

Reviewer #2: (No Response)

7. PLOS authors have the option to publish the peer review history of their article (what does this mean?). If published, this will include your full peer review and any attached files.

Reviewer #1: No

Reviewer #2: **Yes: **Dr Tawakalit Olubukola Salam

---

## [Editor Report · Acceptance letter]

19 Aug 2024

PONE-D-24-14998R1 

PLOS ONE

Dear Dr. Okyere, 

I'm pleased to inform you that your manuscript has been deemed suitable for publication in PLOS ONE. Congratulations! Your manuscript is now being handed over to our production team.

Kind regards, 

on behalf of

Dr. Tijani Idris Ahmad Oseni 

Academic Editor

PLOS ONE